# Evolution of Attenuation and Risk of Reversal in Peste des Petits Ruminants Vaccine Strain Nigeria 75/1

**DOI:** 10.3390/v11080724

**Published:** 2019-08-07

**Authors:** Roger-junior Eloiflin, Marie Boyer, Olivier Kwiatek, Samia Guendouz, Etienne Loire, Renata Servan de Almeida, Geneviève Libeau, Arnaud Bataille

**Affiliations:** 1CIRAD, UMR ASTRE, F-34398 Montpellier, France; 2ASTRE, Univ Montpellier, CIRAD, INRA, Montpellier, France

**Keywords:** PPR, morbillivirus, attenuation, virulence, wild strain, vaccine, deep sequencing

## Abstract

Peste des Petits Ruminants (PPR) is a highly infectious disease caused by a virus of the *Morbillivirus* genus. The current PPR eradication effort relies mainly on the implementation of massive vaccination campaigns. One of the most widely used PPR vaccines is the Nigeria 75/1 strain obtained after attenuation by 75 serial passages of the wild type isolate in cell cultures. Here we use high throughput deep sequencing of the historical passages that led to the Nigeria 75/1 attenuated strain to understand the evolution of PPRV attenuation and to assess the risk of reversal in different cell types. Comparison of the consensus sequences of the wild type and vaccine strains showed that only 18 fixed mutations separate the two strains. At the earliest attenuation passage at our disposal (passage 47), 12 out of the 18 mutations were already present at a frequency of 100%. Low-frequency variants were identified along the genome in all passages. Sequencing of passages after the vaccine strain showed evidence of genetic drift during cell passages, especially in cells expressing the SLAM receptor targeted by PPRV. However, 15 out of the 18 mutations related to attenuation remained fixed in the population. In vitro experiments suggest that one mutation in the leader region of the PPRV genome affects virus replication. Our results suggest that only a few mutations can have a serious impact on the pathogenicity of PPRV. Risk of reversion to virulence of the attenuated PPRV strain Nigeria 75/1 during serial passages in cell cultures seems low but limiting the number of passages during vaccine production is recommended.

## 1. Introduction

Peste des Petits Ruminants (PPR) is a highly infectious disease caused by a virus of the *Morbillivirus* genus, within the *Paramyxoviridae* family [1]. Small ruminants (sheep and goats) are the main hosts of PPR. PPR is an important disease burden and threatens the livelihoods and food security of smallholder farmers across Africa, the Middle East, and Asia [2]. Because of its impact, the FAO and OIE launched a campaign for the global eradication of PPR [3].

The virus causing PPR is a non-segmented RNA virus of the genus *Morbillivirus* in the family *Paramyxoviridae*. Recently, the International Committee on Taxonomy of Viruses (ICTV) has changed the name of the virus from *peste des petits ruminants virus* to *Small ruminant morbillivirus* [4]. Here, the abbreviation PPRV will be used throughout the text for the PPR virus to avoid confusing non-specialist readers interested in the global PPR eradication campaign.

PPRV is an enveloped virus with a single negative RNA strand encoding six structural proteins: nucleoproteins (N), phosphoproteins (P), matrix proteins (M), fusion proteins (F), hemagglutinin proteins (H) and large proteins (L). The P gene also encodes two non-structural proteins, V and C. The N protein encapsidates the viral genome and forms a helical nucleocapsid which interacts with the P and L proteins (RNA-dependent RNA polymerase) to form the ribonucleoprotein (RNP) complex [5]. The two membrane glycoproteins F and H are involved respectively in membrane fusion and in binding the virus to its cell receptor during infection. The M protein appears to play a complex role in virus assembly and is also necessary for infection during which the matrix protein dissociates from the viral membrane for the F glycoprotein to become fusogenic and the nucleocapsid to be released into the host’s cytoplasm [6]. 

PPR eradication efforts rely mainly on massive vaccination campaigns. Like for other morbilliviruses such as measles and rinderpest virus, PPRV vaccines are the product of the successful attenuation of wild virus strains [7,8]. One of the most widely used PPR vaccines is the Nigeria 75/1 strain obtained after attenuation by repeated passages in cell cultures of the wild-type isolate collected from sick sheep and goats during PPR outbreaks in Nigeria [9]. Seventy-five cell passages were performed on Vero cells to produce the attenuated vaccine strain. To assess residual virulence during the attenuation process, susceptible goats were infected with intermediate passages. Infection with the 20th passage still induced mild and transient hyperthermia for 48 h, but no other clinical signs. The 55th, 58th, and 61st passages induced no clinical signs but were able to confer protection against additional experimental infection with PPRV [10]. Additional passages were necessary to obtain a purified clone to be used for vaccine production (Adama Diallo, personal communication).

The mechanisms and factors involved in the attenuation of morbilliviruses remain poorly known. Attenuation processes generally involve the occurrence and selection of several mutations along the genome. Mutation rates and selection pressure depend on the virus strain and on the cells used for isolation and attenuation.

In the measles virus (MeV), another morbillivirus, it has been shown that a strain isolated from the same patient with acute measles, whether isolated in B95a cells or on Vero cells, induced typical measles or no clinical signs at all, respectively, in infected monkeys. Both isolates differed in only two nucleotide mutations, one on the P/V/C gene and the other on the M gene. In this case, these differences appeared to be responsible for discrepancies in the cell tropism and pathogenicity of MeV [11,12].

Better assessment of the evolution of attenuation and of viral populations in cell cultures is especially important for PPRV, as it may help identify specific mutations or genome positions which influence the virulence/attenuation of PPRV. The safety and efficacy of the Nigeria 75/1 vaccine strain is already well characterized [13], and its genome has been fully sequenced, notably using deep sequencing in the framework of quality control of the master seed held by CIRAD (Sequence Read Archive, accession number SRP100964). Currently, more than 20 government-backed and private companies produce the Nigeria 75/1 vaccine. Producers are typically advised not to go further than five cell passages of the vaccine seed during production to avoid risks of reduced vaccine efficacy and of eventual reversal to virulence [14]. However, limited information is available on whether these guidelines are followed or even adopted. 

Risk of reversion to virulence is poorly studied in morbilliviruses, maybe because so far, there have been no reports from the field of potential reversion to virulence of PPR, rinderpest or measles vaccines. However, due to the genetic plasticity of the RNA viruses, it would be possible for mutations associated with a reversion to virulence to be generated in a viral population. Notably, work with live attenuated vaccines of other viruses, such as Rift Valley Fever virus, show that genetic drift during cell passage may present a risk of reversal [15]. 

High throughput sequencing is a powerful tool to explore virus genetic diversity, and therefore, to monitor the emergence and fixation of new variants over time. In particular, the technique has been used to study intra-host diversity of RNA viruses, including West Nile, Newcastle Disease, Chikungunya, and Dengue fever viruses [16,17,18,19]. The technique could be used to monitor the evolution of genetic diversity in viral population in vitro during the attenuation process and could help identify specific mutations and genome positions which influence the virulence/attenuation of PPRV. CIRAD holds the original wild type strain at the origin of the vaccine Nigeria 75/1, some of the historical cell culture passages that led to the attenuation of this strain, and also up to the 140 passages produced after the vaccine. These passages offer a unique opportunity to explore the evolution of attenuation of the PPR vaccine strain Nigeria 75/1, its genetic plasticity and risk of virulence reversion, over cell passages.

In this study, we applied high throughput sequencing to the different cell passages of PPRV during and after attenuation of the Nigeria 75/1 strain in different cell types. Additionally, we used intermediate PPRV passages to infect a murine cell line known to be differentially affected by virulent and attenuated PPRV strains [20]. The immunofluorescence test was used to analyze the level of infection at different passages in the murine cell line to help identify mutations with a putative role in virus attenuation. 

## 2. Material and Methods

### 2.1. Viral Strains

Viral strains available at CIRAD include the wild type Nigeria 75/1 strain, and intermediate passages 47 to 75 (passage 75 corresponds to the vaccine master seed). In the case of P47, titer was extremely low, and an additional passage on CV1 cells expressing the goat-sheep Signaling Lymphocyte Activation Molecule SLAM (CHS cells) [21] was needed to recover a culture with a sufficiently high titer to allow genome sequencing and infection studies.

During the development of the PPR Nigeria 75/1 vaccine, the viral strain, attenuated after 75 passages in cell cultures, underwent additional passages (from passage 75 up to passage 140). CIRAD also stores some of these posterior passages (P89, P90, P99, P130, P149). However, the passages directly following the vaccine strain (passages 76 to 85) were not available. To test the genetic stability of the vaccine in the first few passages in a different type of cells, 10 additional passages were performed starting from the 75th passage on two types of cells in parallel: Vero cells and CHS cells. 

### 2.2. Cell Culture Passages

The Vero and CHS cells cultured in a MEM medium (minimum essential medium Eagle). C3H/10T1/2 (10T1/2) cells, a mouse fibroblastic embryonic cell line (ATCC CCL-226, Manassas, Virginia, USA) were cultured in basal medium Eagle (BME). All media were supplemented with 2 mM L-Glutamine and 10% fetal bovine serum.

The additional passages of the vaccine master seed from 76 to 85 were obtained by sequential infection from the previous passages on Vero and CHS cells. Virus solutions were first titrated on Vero cells using the method of Spearman–Karber [22]. Infection was carried out directly in 96-well plates. After a freezing-thawing procedure, supernatants from virus-infected wells were collected and titrated. The titrated cells were directly used for infection of the following cell passages.

### 2.3. High Throughput Genome Sequencing of PPR Virus Populations

Viral RNA extraction was performed with a KingFisher^TM^ mL extraction robot and the ID kit Gene™ Mag Universal Extraction (IDvet, Grabels, France), following the manufacturer’s instructions. cDNA synthesis and PCR amplification of the complete PPRV genome was performed on five overlapping fragments: F1 (4050 pb), F2 (3663 pb), F3 (3816 pb), F4 (3800 pb) and F5 (3335 pb). The details of the primers used are provided in Appendix A. cDNA synthesis was performed with a Reverse Transcriptase (RT) (RevertAid RT kit, Thermo Scientific, Courtaboeuf, France) and the specific primers of each fragment (F1 to F5), following the instructions provided with the kit. The RT-PCR program was set up as follows: 65 °C for 5 min, 42 °C for 60 min and 70 °C for 5 min. First-strand cDNA of each fragment was then amplified by PCR according to the kit protocol KAPA DNA Polymerase 250 unit (KAPA HiFi PCR Kit Kapa Biosystems, Champagne-au-Mont-d’Or, France). PCR of fragment 1 was performed with the following program: 95 °C for 3 min, 10 amplification cycles (20 s at 98 °C, 15 s at 55 °C and 4 min at 72 °C), 25 amplification cycles (20 s at 98 °C, 15 s at 58.9 °C and 4 min at 72 °C) and a final extension step at 72 °C for 4 min. The PCR program for the other fragments was 95 °C for 3 min, 10 amplification cycles (20 s at 98 °C, 15 s at 50 °C and 4 min at 72 °C), 25 amplification cycles (20 s at 98 °C, 15 s at 55.2 °C and 4 min at 72 °C) and a final extension step at 72 °C for 4 min. The PCR products were then separated by electrophoresis on 1.5% agarose gel and the fragments of the expected sizes excised and purified using the NucleoSpin^®^ Gel and PCR Clean-up kit purification kit following the manufacturer’s instructions (Machery-Nagel, Hoerdt, France). The different amplified fragments were quantified using a Qubit fluorometer (Qubit^®^ 2.0 fluorometer, Thermo Scientific, Courtaboeuf, France) and pooled in a single tube in an equimolar manner. The pool was used to prepare sequencing libraries.

For the wild type Nigeria 75/1, library preparation and sequencing had been done before this study using the HiSeq Illumina platform (Macrogen, South Korea). For the vaccine strain, deep-sequencing data had already been obtained in our laboratory (Kwiatek, personal communication). For all the other strains, the protocol below was followed. The vaccine strain was sequenced again with the following protocol. 

Samples were diluted to 0.2 ng/µL, and library preparation was performed according to the instructions provided in the Nextera kit (Illumina). The libraries were prepared in “dual index”, i.e., using two indexes for each sample.

The quantitative PCR (Q-PCR) “Library quantification kit” (Takara Clonetech, St-Germain-en-Laye. France) was performed according to the manufacturer’s instructions on a LightCycler instrument (Roche, France) to assess the exact quantity of material for each library. The distribution of fragment sizes within each library was analyzed on an Agilent 2100 bioanalyzer with the Agilent high sensitivity DNA kit (Agilent Technologies, Les Ulis, France), according to the manufacturer’s instructions. The quantity and average size of each library were used to pool the libraries in an equimolar manner. 

Sequencing was done on a MiSeq Illumina machine at the AGAP sequencing platform (CIRAD, Montpellier, France). The samples were pooled in a group of maximum eight samples to increase the chances of obtaining a minimum depth of 100× across the entire PPR genome to study genetic diversity within viral populations.

The extremities of the genomes are usually very difficult to obtain by NGS sequencing because only a few or no good quality reads are obtained in these regions. To complete the extremities of the genome, RACE PCR (rapid amplification of cDNA-ends by polymerase chain reaction) was performed following the instructions in the 5′/3′ RACE kit second-generation (Roche, France) [23]. The PCR products obtained were sequenced using classical Sanger method at Genewiz (Bishop’s Stortford, United Kingdom).

### 2.4. Bioinformatic Analyses

At the end of the Illumina sequencing, the raw data obtained were first analyzed using a homemade bioinformatics pipeline (https://github.com/loire/Varhap) to determine the consensus sequence within the sample, as well as to detect nucleotide variations (SNVs) within the population. The pipeline calls for modules available on SAMtools [24] for the different steps of the analysis, including alignment of reads (BWA) and variant calling freebayes [25]. Variant Call Format (VCF) files summarised the exact proportion of each variant detected. These variants were determined by applying a filter on raw data. The filter most frequently used to detect variants in this study was (vcffilter -f “QA/AO > 10 & AO/RO > 0.01 & DP > 100 & RPL > 2 & RPR > 2 & SAF > 2 & SAR > 2”) where QA represents the addition of base? “A” qualities, A0 (A for Alternative) represents the number of reads on the alternative base, R0 (R for reference) represents the number of reads on the reference base, DP represents the depth, RPR and RPL reads “balanced” to each side (left and right) of the site. SAF/SAR represent the number of reads on the forward/reverse sequences. 

For the wild type Nigeria strain, the genome of the vaccine strain Nigeria 75/1 (GenBank Accession n° KY628761) was first used as a reference. For all other passages, the genome of the wild type Nigeria strain was used as a reference to highlight evolution towards attenuation. The analysis was performed on the SouthGreen computing cluster (CIRAD, Montpellier, France). Geneious R11 and IGV (Integrative Genome Viewer) software were used to visualize variants and to double-check alignment quality. 

The genetic distance between the consensus sequence pairs obtained was calculated using the Manhattan distance method. The Manhattan distance between the viral population and consensus of the same sample was used as a measure of intra-sample genetic diversity [26]. A matrix was constructed where the part below the diagonal contains the inter-sample Manhattan distances between pairs of consensus sequences, the part above the diagonal contains the inter-sample Manhattan distances between the frequencies of the variants, and the diagonal represents the intra-sample Manhattan distances between the consensus sequence and the frequency of the variants. The different distance matrices obtained were used to visualize in two dimensions the genetic difference between strains following Höper et al. (2015) using the package Classical (Metric) Multidimensional Scaling (cmdscale) in R Studio version 1.1.456.

For the wild type, data on variant frequencies were not used for the analyses, as different sequencing protocols had been used and the results concerning genetic diversity were, therefore, not comparable. 

All genomes obtained were aligned to PPRV genomes publicly available on GenBank using Clustal W as implemented in the Bioedit software to assess nucleotide variability across PPRV strains. 

### 2.5. Immunofluorescence on 10T1/2 Cells

The 10T1/2 cell line was used in this study to evaluate the virulence status of different PPRV passages. Indeed, this cell line has previously shown different infection phenotype when infected by the PPRV vaccine strain or wild Nigeria 75/1 strain [20]. Virus infection was observed using immunofluorescence based on antibodies targeting the viral N protein [27]. Viral suspensions were first titrated using the method of Spearman–Karber. A 96-well flat-bottom plate was seeded with 2.10^4^ cells per well. Different wells were infected with different viral suspensions with a multiplicity of infection (MOI) between 0.1 and 0.001 (depending on the different titers obtained). Parallel infections were carried out on Vero cells. Once the cytopathogenic effect (CPE) reached 50–60% in the Vero cells, infection of both cell types was stopped by removing the culture medium and adding an ice-cold 80% acetone solution to permeabilize cell membranes after which the cells were incubated at −20 °C for 30 min. Acetone was removed and the cells were washed three times with PBS 1X to remove excess acetone. N-antibody coupled with TRITC fluorochrome was added at a dilution of 1:100. The cells were incubated at 37 °C for 30 min and then washed three times with PBS 1X. A volume of 100 µl of PBS was then added to each well before the cells were observed using a fluorescence microscope.

## 3. Results 

### 3.1. Comparison of Consensus Genome between PPRV Wild Type and Vaccine Nigeria 75/1 Strain

Sequencing coverage and the depth obtained for each genome are shown in Figure 1 (sequencing read archive accession number XXXX, GenBank accession numbers: xxxx-yyyy). Except for the wild type strain, RACE amplification was necessary to obtain the last 1–10 bp of each extremity of the genome. Comparison of the consensus sequences for the wild type and vaccine Nigeria 75/1 strains showed that only 18 mutations differentiate the two strains. Seventeen out of 18 mutations were present at a frequency of 100% in sites with above 100x coverage, while the wild type variant at position 36 was still present at 14% frequency in the vaccine strain (Table 1). These mutations were distributed throughout the genome both in coding and non-coding regions and in the leader and trailer regions of the genome (Table 1). Nine mutations were synonymous and nine were not.

Non-synonymous mutations were observed at positions: 1513 (L => P), 3158 (G => E), 3694 (K => R), 4046 (R => S), 6422 (K => E), 7384 (N => T), 8829 (R => S), 12193 (D => A) and 15777 (D => N). All this information is summarized in Table 1.

Twelve out of the 18 mutations observed are at genome positions that are not variable in other PPRV genome strains available publicly (see Appendix A). The vaccine variants at position 5 and 1513 are shared with another vaccine strain (Sungri 1996). Positions 36, 221, 4046, and 4559 are variable in multiple PPRV genome included in the alignment (see Appendix A). 

### 3.2. Appearance of Variants in Intermediate Attenuation Passages

The distribution of single nucleotide polymorphisms (SNP) along the genomes is shown in Figure 2. Variants were found throughout the genome, although fewer were observed along the L viral protein.

At the earliest passage at our disposal (Passage 47 on Vero cells + one passage on CHS cells), 12 out of the 18 mutations were present at a frequency of 100% (Table 1). However, a variant was identified at a low frequency (2–3%) at position 36 in most passages after passage 47. Five of the remaining six mutations appeared between P47 and P62. Three of these mutations were already present at a frequency of 100% at P62, whereas the “vaccine” variant reached 100% frequency at P65 for position 36 and at P69 for position 8829. RACE PCR and Sanger sequencing of the PPRV genome leader and trailer regions showed that the mutation at position 15,946 appeared in the population at passage 58 on Vero cells. Both vaccine and wild type variants were observed on the chromatogram at position 5 in the leader region of PPRV genome for P65. Only the vaccine variant was observed at P67 at the same position (Appendix A). 

### 3.3. Appearance of Genetic Variants in Passages after the Vaccine Master Seed

All 18 mutations differentiating the wild type and vaccine strains remained fixed in the population with no change in frequency except for mutations at positions 36, 4046 and 5627 at which the wild type variant sometimes reappeared at low frequencies (Table 2). Low-frequency variants were observed in other regions of the genome in these passages. The distribution of low-frequency variants was similar to that observed in intermediate attenuation passages, except that more variants appeared in the L gene at P90, P99, P130, and P140 (Figure 2). Some new variants increased in frequency and reached 100% in passages after the vaccine strain (Appendix A).

Two types of cells (Vero or CHS) were used to produce up to 10 new passages from the vaccine strain. Comparison of viral genetic diversity within the two types of cells (Vero or CHS) revealed a strong increase in the amount of SNPs in the 5th passage produced on CHS (called Vero75 CHS5) compared to the 5th passage produced on Vero cells (called Vero80, 81 SNPs in Vero 80 versus 394 SNPs in Vero 75 CHS 5). However, after 10 passages, the same amount of SNPs was observed in both types of cells (Vero 85 = 366 SNPs, Vero 75 CHS 10 = 377). NMDS analyses showed that the evolution of passages after the vaccine strain differed between historical passages (P90, P99, P130, P140) and the newly produced passages (P80, P85, V75C5, V75P10, Figure 3). When only the consensus sequences were used to calculate genetic distances, the distribution of strains in the graph coincided with sequential cell passages (Figure 3A). When all the variants in each population were taken into account to compute genetic distances, the distribution appeared to be more complex (Figure 3B). The genetic differentiation of V75C5 and V75C10 from other passages was more apparent. Genetic diversity appeared especially high in P70, isolating this passage from previous or subsequent passages. P90 was in an intermediate position between wild type and vaccine strains.

### 3.4. In Vitro Infection of 10T1/2 Cell Line

In this cellular model, immunofluorescence tests revealed a large number of viral particles in cells infected by the wild type Nigeria strain, and in all intermediate passages up to passage 65. No or few viral particles were detected in cells infected with the vaccine strain, or in passages 67 and beyond (Figure 4). 

## 4. Discussion

Mechanisms leading to attenuation in morbilliviruses are poorly understood. Many studies have been conducted on the attenuation of the measles virus, but all focused on the final formulation of the vaccine and did not investigate the process that led to the attenuation process [8,12]. High throughput sequencing of the original wild type strain and attenuation passages that led to vaccine Nigeria 75/1 strain offers a unique opportunity to identify mutations that are potentially important for PPRV attenuation and to better understand the evolution of morbillivirus during serial passages in cell cultures.

Our results show that only 18 mutations separate the wild type strain and the vaccine strain. Even if the wild type strain was not highly pathogenic, it still provoked clinical symptoms and could be transmitted [10]. Therefore, some of these mutations may play an important role in determining PPRV pathogenicity. 

Thirteen mutations were located in coding regions of the viral proteins, of which nine were non-synonymous mutations. Some of these mutations may modify important protein-protein interactions. Mutations in glycoproteins F and H may affect the capacity of the viral particle to bind to cell receptors, and therefore, could directly affect the virus entry mediated by F protein and triggered by an interaction with H protein [11]. As observed in the measles virus, adaptation of PPRV to Vero cell culture may procure the ability to use the complement regulator CD46 as a major receptor to mediate virus entry and intercellular fusion [28]. 

Unlike virulent strains, PPRV vaccine strains are only lymphotropic and no longer epitheliotropic, impeding virus shedding, and transmission between animals [10]. Mutations in the coding sequence of P/V/C and M proteins may modify cell tropism and pathogenicity of the vaccine strain. Notably, comparative nucleotide sequence analyses revealed that only two nucleotide differences separated two measles virus strains with different cell tropism and pathogenicity, one in the P/V/C gene and the other in the M gene [11]. 

Mutations in the L and P coding sequence may induce dysfunctional activity of the polymerase and variable efficiency of transcription, which can directly affect protein expression [29]. The L gene of an attenuated MeV strain has also been shown to reduce gene expression and virus propagation in cell culture and in mice expressing hSLAM [30]. Therefore, mutations in the polymerase protein in PPRV could contribute to an attenuated phenotype, but also to the stability of attenuated phenotype and the increase in viral replication [31].

Synonymous mutations and mutations in non-coding regions may also be important for the Nigeria 75/1 attenuation process. Synonymous mutations may lead to codon deoptimization, a process linked to attenuation of several viruses including Zika and influenza A viruses [32,33]. Sequences of nine vaccine MeV strains compared with Edmonston wild-type measles virus sequence revealed differences in the noncoding regions of the genomes as well as in coding regions in all viral proteins [12]. The 5′/3′ UTR regions are involved in the fixation of the transcription factor for the RNA-dependent viral polymerase. All cis-acting regulatory elements such as promoters and encapsidation signals are contained in the first 107 and the last 109 nucleotides of the genome [34]. Consequently, mutations detected in the leader and trailer regions may interfere with viral replication. The mutation observed in the inter-regions between M and F could affect PPRV replication and cytopathogenicity. By having both long M and F UTRs, MeV may replicate efficiently and minimize cytopathogenicity [35]. Moreover, regions rich in GC could fold in complex secondary RNA structures and play a role in regulating transcription of the M and F genes [12]. 

Taken together, these studies of MeV suggest that all 18 mutations observed between virulent and attenuated PPRV Nigeria 75/1 strains could play a role in its attenuation. Our study provides only a few clues to whether some mutations are more important than others. Indeed, 12 out of 18 mutations were already fixed in the viral population at passage 47. According to the results of the original attenuation experiments [10], after 20 passages in cell cultures, Nigeria 75/1 strain provoked only a transitory rise in temperature over a period of 48 h following the incubation period (5–6 days) when inoculated in goats. From the 55th passage on, infected animals presented no clinical signs of the infection, and were unable to infect naïve goats through direct contact. Therefore, the key mutations for attenuation of the wild type strain should be among the 12 mutations that appeared before passage 55. Sequencing of the passages before passage 47 would be necessary to identify the time of appearance of the 12 mutations during the attenuation process and perhaps to be able to link some of them with the loss of symptoms in goats.

Results of in vitro infections of 10T1/2 cells suggested that the mutation at position 5 in the leader region of the PPRV genome is very important for the capacity of virus replication in this cell type. It is not clear if this mutation is also important for in vivo virulence. However, comparisons of PPRV genomes available in GenBank show that, to date, this mutation has only been found in vaccine strains and in one strain isolated in Ethiopia in 1994 (GenBank accession number: KJ867542). This mutation is present in other paramyxoviruses, either in vaccines or virulent strains [36,37]. In addition, IFIT-1, an interferon-induced antiviral effector, can bind to the 5′ UTR region. Mutations in the 5′ UTR have been reported to block recognition of IFT-1, and therefore, could disrupt the function of this antiviral protein [38]. Conversely, the attenuation process may involve mutations in the 5′ UTR, which could facilitate IFIT-1 binding and immune restriction. 

Unfortunately, it was only possible to obtain the extremity of the genomes by Sanger sequencing, so the importance of low frequency variant could not be fully explored. Still, the presence of the two variants was identified in passage 65, a passage still able to infect cells 10T1/2, suggesting that the appearance of the vaccine variant did not impede the capacity of some of the viral population to infect these cells. Further in vitro and in vivo experiments are needed to explore the role and importance of this mutation.

Our results also identified some diversity of variants in each virus population. Low-frequency variants appeared to be distributed along the whole genome in a consistent manner across passages. Our results showed no low-frequency variants at the positions of the 18 mutations potentially associated with attenuation, except position 36 and 4046. These two positions are variable across PPRV lineages, suggesting that the presence or absence of specific variant at these positions is not important for the attenuation of PPRV. In some instances, some variants increased in frequency and became fixed in the population (for example T1359A, fixed after the passage 130, Appendix A), suggesting selection pressure possibly favoring adaptation to the cell culture system [28]. Results from passages performed in parallel on Vero and CHS cells show that diversity is higher in CHS cells, especially after five passages. Notably, a wild type variant at one of the 18 mutations potentially associated with attenuation (position 5627) re-appeared at low frequency. These results suggest that the virus is subject to different selection pressure in different types of cells, with CHS cells representing a higher risk of reversal or loss of vaccine efficacy, possibly due to the expression of SLAM receptors in these cells. This hypothesis is supported by results of a previous infection study showing that cell culture passages affect the pathogenicity of PPRV strains, with strains passaged in cells expressing SLAM demonstrating stronger virulence [39].

Multidimensional analysis showed that the new passages produced were genetically different from historical passages after the vaccine strain. This may be due to differences in the cell culture and in the passage methodology used, resulting in a different virus replication rate, and therefore, different viral population diversity and selection pressures in the system. Therefore, the pathway to attenuation or to evolution in cell cultures, in general, may differ even for the same PPRV strain depending on the method used. For example, in another attempt to produce vaccine from the same wild-type strain, loss of clinical symptoms of PPRV was not observed before the 65th passage [40] much later than the 20th passage in the first study [10]. 

Our study suggests that only a few mutations can have a serious impact on the pathogenicity of PPRV. Further work on the effect of these mutations on virus replication or protein-protein interactions may provide advance our understanding of the mechanisms of attenuation in morbilliviruses. Moreover, information on the mutations involved in attenuation may help design live-attenuated viruses, notably through genome re-encoding [41]. Importantly, our results show that the risk of reversal of PPRV strain Nigeria 75/1 after serial passages in Vero, and even CHS cell cultures, seems low. However, it is still recommended to limit the number of passages during vaccine production to ensure the quality of vaccines to be used in the eradication campaign. 

## Figures and Tables

**Figure 1 viruses-11-00724-f001:**
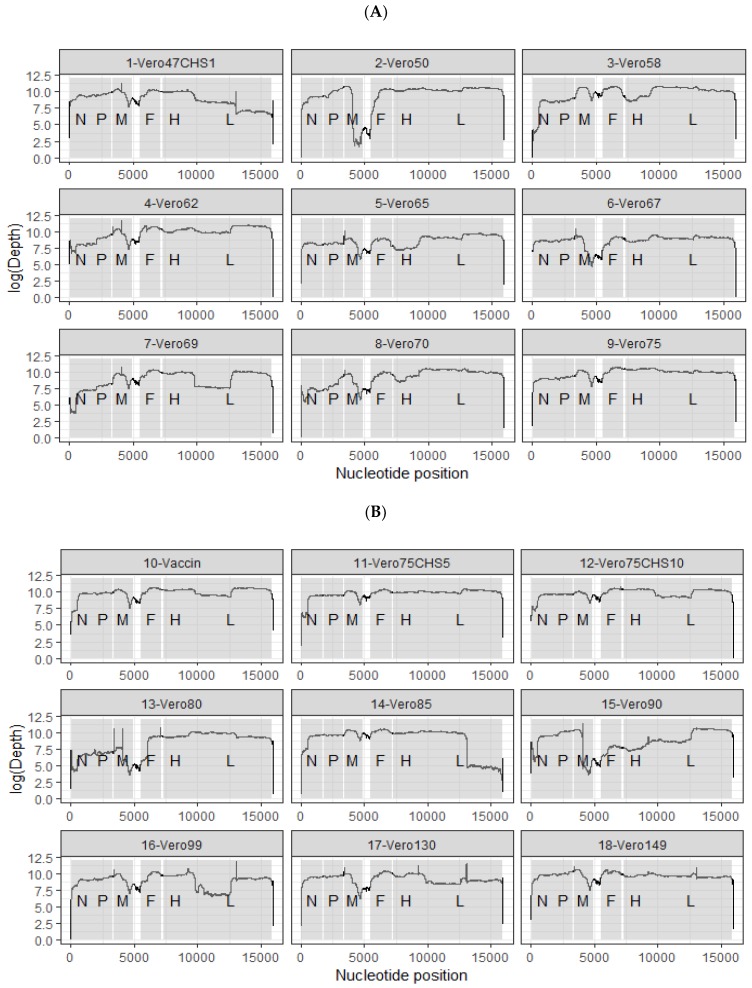
Summary of the different viral strains made available during this study. Coverage and depth of all the sequenced passages are shown in both (**A**) and (**B**). CHS: CV1 cell line stably expressing the goat SLAM.

**Figure 2 viruses-11-00724-f002:**
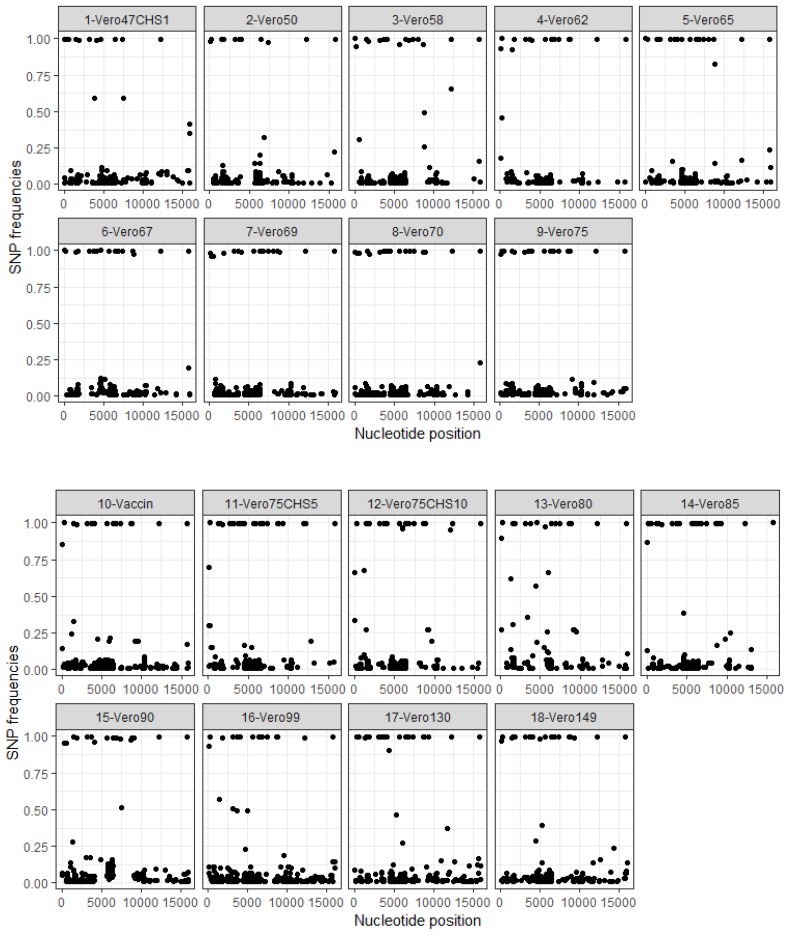
Distribution of SNP frequencies according to positions in the genome. SNP: Single nucleotide polymorphism. Position: PPRV genome from 0 to 15,948 bp. The Vero47CHS1 passage represents 47 passages on the Vero cells and one passage on the CHS cells. The Vero75CHS5 passage represents 75 passages on Vero cells and five passages on CHS cells.

**Figure 3 viruses-11-00724-f003:**
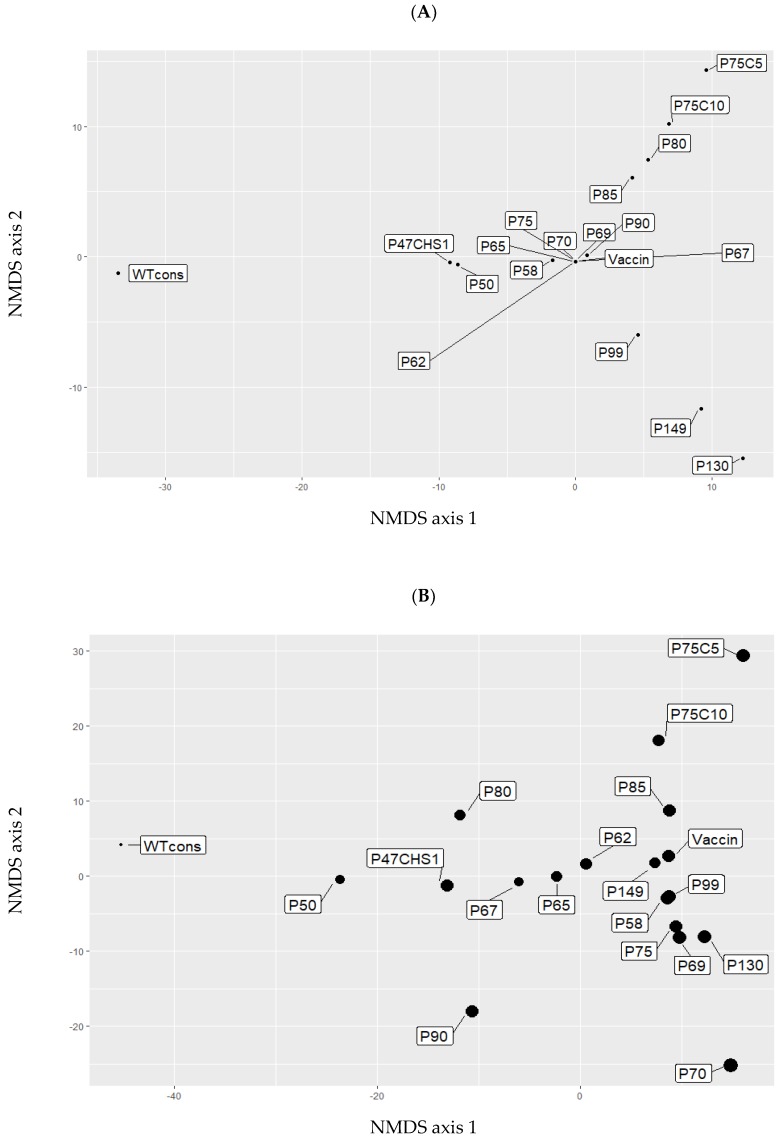
Non-metric multidimensional scaling (NMDS) between each sequenced passage. The obtained NMDS was calculated from matrix (Appendix A). A: NMDS of consensus sequence, B: NMDS of intraspecific diversity. WT cons: Wild type consensus. P47CHS1: Passage 47 on Vero cells + one passage on CHS cells. P50 50th passage on Vero cells (like other P… in these plots).

**Figure 4 viruses-11-00724-f004:**
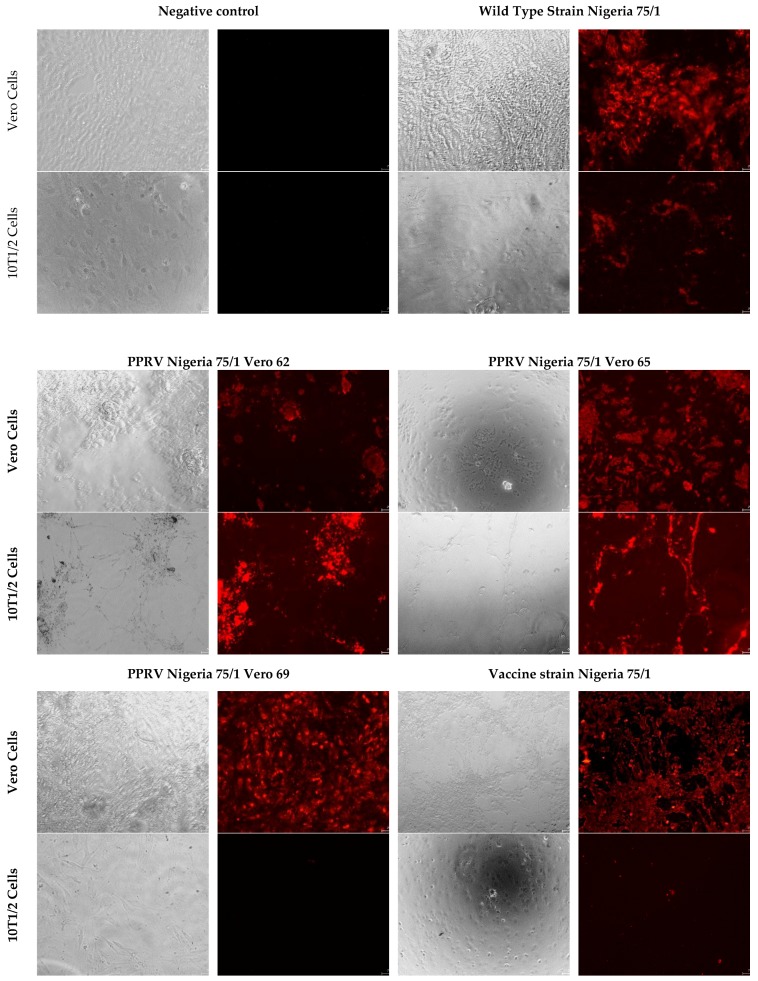
Infection of 10 T1/2 cells with different cellular passages leading to the attenuation of the vaccine strain Nigeria 75/1.

**Table 1 viruses-11-00724-t001:** Evolution of the mutations that led to mitigation of the vaccine strain before the vaccine was obtained.

Mutations	Regions	POS	Passages										
			Wild Type	Vero 47 CHS1	Vero 50	Vero 58	Vero 62	Vero 65	Vero 67	Vero 69	Vero 70	Vero 75	Vaccin CIRAD
1	Leader	5*	A	.	.	.	.	A/G	A => G	.	.	.	.
2	Leader	36	A	A => G (100%)	.	.	A/G (3%)(97%)	A/G (0%)(100%)	.	A/G (2%)(98%)	A/G (100%)	A/G (2%)(97%)	A/G (14%) (85%)
3	N CDS	221	T	T => C (100%)	.	.	.	.	.	.	.	.	.
4	N CDS	1513	T	T => C (100%) L => P	.	.	.	.	.	.	.	.	.
5	P mRNA	1795	C	C => T (99%)	.	.	.	.	.	.	.	.	.
6	P CDS	3158	G	G => A (99%) G => E	.	.	.	.	.	.	.	.	.
7	M CDS	3694	A	A => G (100%) K => R	.	.	.	.	.	.	.	.	.
8	M CDS	4046	A	A => C (1%) (99%) R => S	.	.	.	.	.	.	.	.	.
9	M mRNA	4559	G	G => A (100%)	.	.	.	.	.	.	.	.	.
10	F CDS	5627	A	.	A => G (14%)(85%)	A/G (4%) (96%)	.	.	.	.	.	.	.
11	F CDS	6422	A	A => C (100%) K => E	.	.	.	.	.	.	.	.	.
12	F CDS	6846	A	.	A/G (68%)(32%)	A => G (1%) (99%)	.	.	.	.	.	.	.
13	H CDS	7384	A	A => C (100%) N => T	A/C (3%) (97%)	.	.	.	.	.	.	.	.
14	H CDS	8645	C	.	C/T (91%) (9%)	C => T (4%) (96%)	.	.	.	.	.	.	.
15	H CDS	8829	C	.	.	C/A (74%) (26%)	C => A (100%) R => S	C/A (17%)(83%)	C/A (3%) (97%)	.	.	.	.
16	L CDS	12193	A	A => C (100%) D => A	.	.	.	.	.	.	.	.	.
17	L CDS	15777	G	G => A (100%) D => N	.	.	.	.	.	.	.	.	.
18	Trailer	15946*	C	/	/	C => G	.	.	.	.	.	.	.

Mutations (1:18) represent the 18 mutations separating the wild type Nigeria strain from the vaccine strain; POS, position in the PPRV genome; Vero x CHS y, virus obtained after x passages in Vero cells and y passages in CHS cells; (/) no data obtained; (.) Change equal to the change observed in the previous passage. Percentages represent the frequency of presence of certain bases within the population. The positions highlighted are those that host non-synonymous mutations. * Results obtained after RACE analysis.

**Table 2 viruses-11-00724-t002:** Evolution of the mutations that led to the mitigation of the vaccine strain after the vaccine was obtained.

Mutations	Regions	POS	Passages								
			Vaccin CIRAD	Vero 75 CHS 5	Vero 75 CHS 10	Vero 80	Vero 85	Vero 90	Vero 99	Vero 130	Vero 149
1	Leader	5	G	/	/	/	/	/	/	/	/
2	Leader	36	A/G (14%) (85%)	A/G (30%) (70%)	A/G (33%) (67%)	A/G (9%) (90%)	A/G (13%) (87%)	A/G (4%)(95%)	A/G (7%) (93%)	A/G (3%) (97%)	.
3	N CDS	221	C	.	.	.	.	.	.	.	.
4	N CDS	1513	C	.	.	.	.	.	.	.	.
5	P mRNA	1795	T	.	.	.	.	.	.	.	.
6	P CDS	3158	A	.	.	.	.	.	.	.	.
7	M CDS	3694	G	.	.	.	.	.	.	.	.
8	M CDS	4046	C	.	.	A/C (8%) (91%)	A/C (100%)	A/C (4%)(96%)	A/C (100%)	.	.
9	M mRNA	4559	A	.	.	.	.	.	.	.	.
10	F CDS	5627	G	.	.	A/G (2%) (98%)	A/G (100%)	.	.	.	.
11	F CDS	6422	C	.	.	.	.	.	.	.	.
12	F CDS	6846	G	.	.	.	.	.	.	.	.
13	H CDS	7384	C	.	.	.	.	.	.	.	.
14	H CDS	8645	T	.	.	.	.	.	.	.	.
15	H CDS	8829	A	.	.	.	.	.	.	.	.
16	L CDS	12193	C	.	.	.	.	.	.	.	.
17	L CDS	15777	A	.	.	.	.	.	.	.	.
18	Trailer	15946	G	/	/	/	/	/	/	/	/

Mutations (1:18) represent the 18 mutations that led to the mitigation; (/) no data obtained; (.) Change equal to change observed in the previous passage. Percentages represent the frequency of presence of certain basis within the population. The positions highlighted are those that host non-synonymous mutations.

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
