# Peer review of "Evolution of Attenuation and Risk of Reversal in Peste des Petits Ruminants Vaccine Strain Nigeria 75/1"

_viruses, 2019, doi:10.3390/v11080724_

Round 1
Reviewer 1 Report
In this manuscript , the authors use predominantly high throughput deep sequencing to identify the mutations arising in the evolution of the Nigeria 75/1 vaccine strain of Peste de petits Ruminants virus, a member of the Morbillivirus genus. In addition to the wild-type virus and the vaccine strain, the authors have in hand a few of the serial tissue culture passages leading to the 75thpassage that constitutes the vaccine. Through these studies, they identify a total of twelve non-synonymous mutations scattered throughout both coding and non-coding regions of the genome. They go on to discuss how each mutation could contribute to the attenuation of the virus, as well as the risk of reversal of the attenuated phenotype. Based on these results, they conclude that relatively few mutations could account for the attenuated phenotype and that the risk of reversal to a wild-type phenotype is expected to be low.
This is an excellent manuscript, considered to be both thorough and complete. For the most part, the authors do an excellent job of evaluating the potential impact of each of the various mutations. This manuscript contributes in important ways to our understanding of PPR vaccine dynamics and has only minor points that should be addressed before the manuscript can be considered acceptable for publication.
First, the authors have apparently submitted the manuscript in a form in which they were apparently still considering different ways to express themselves. See lines 180, 185, 199-200, where they have left the apparently alternative choices complete with question marks in the text.
Second, in lines 287-288, the statement that mutations in the H and F proteins could affect receptor binding fails to mention that these mutations could also directly affect virus entry mediated by F, possibly triggered by an interaction with H. Please add this to the discussion.
Author Response
1.First, the authors have apparently submitted the manuscript in a form in which they were apparently still considering different ways to express themselves. See lines 180, 185, 199-200, where they have left the apparently alternative choices complete with question marks in the text.
ANSWER: All alternative choices and question marks have been removed from the text.
2.Second, in lines 287-288, the statement that mutations in the H and F proteins could affect receptor binding fails to mention that these mutations could also directly affect virus entry mediated by F, possibly triggered by an interaction with H. Please add this to the discussion.
ANSWER: This additional information has been added to the discussion.
Reviewer 2 Report
This paper describes amino acid sequence variations in the Small ruminant morbillivirus strain Nigeria 75/1, which is used for producing vaccines against the Peste des Petits Ruminants, one of the most relevant disease of small ruminants in developing countries and one of the major emerging risk for the European countries. The authors describe synonymous and non-synonymous mutations in sequences of viruses after different number of passages on two different types of cell: Vero and CHS cells.
The article is interesting and adds depth to the present knowledge. The manuscript is well written, the research designs is appropriate and conclusions are supported by the results, which are clearly presented. Although I am not competent for revising the English, it seems that the English is correct. I suggest to accept this paper and I suggest only few minor revisions.
actual taxonomy of viruses should be revised; line 8 “Morbillivirus” should be written in italics; line 31 “sometimes called small ruminant morbillivirus” I suggest to report that since 2016 peste-des-petits-ruminants virus (PPRV) has been renamed as Small ruminant morbillivirus (in italics), more information at https://talk.ictvonline.org//taxonomy/p/taxonomy-history?taxnode_id=201851619; line 57 “Measles virus” had this name up to 2015. Since 2016 it has been renamed as Measles morbillivirus. Please correct. More information at https://talk.ictvonline.org//taxonomy/p/taxonomy-history?taxnode_id=201851616
in the text there are many double spaces among consecutive words and some typing mistakes are present, e.g. Line 12: “strain, please check if “,” is needed; line 28: “PPP” probably should be replaced with PPR, line 62: “[11][12]” should be [11,12], line 66: “characterized[13]” a space should be inserted; line 93: “strains[19]” a space should be inserted; line 112: “cell line, (ATCC CCL-226, USA)” should be “cell line (ATCC CCL-226, USA),”; line 138 “France).The” a space should be added after point; line 180: “highlight? Evolution” is “?” correct? Line 182: “was” or “were”?; line 199: “phenotype?” is correct?
line 71: “Producers are typically advised not to go further than five cell passages of the vaccine seed during production to avoid risks of reduced vaccine efficacy and of eventual reversal to virulence.” if this sentence is supported by references, please insert them.
Line 105: I suggest to replace “(P89, 90, 99,130,149)” with (P89, P90, P99, P130, P149)
Line 106: please check if “(passage 76 to 85)” should be changed with (passages 76 to 85)
Table 1 and Table 2: I suggest to mark the non-synonymus mutations that are described at lines 222-223.
Line 201: Please, could you add more information on the antibodies? Were they commercial? Please, insert references.
Line 247: I suggest to write P before each number or to write "at passages" and to write only numbers
Figure 3: Please insert a title to each x- and y-axis
Figure 3: “(like other Ps… in these plots).” “Ps” is not clear
Line 271: is the correct number Figure 4 instead of Figure 3?
Line 273 “morbilliviruses” italics is not required
Line 294: (MeV) has already been defined at line 57, please delete
Line 346-348: “Are there some reference supporting this sentence? Is it an observation done by the authors? In this latter case, I suggest to add these parts in the material and methods and in the results sections.
Line 391: please insert details to reference number 3 (e.g. Internet web site address).
I try to attach a pdf file where suggestions are marked.
Kind regards.

Author Response
actual taxonomy of viruses should be revised; line 8 “Morbillivirus” should be written in italics; line 31 “sometimes called small ruminant morbillivirus” I suggest to report that since 2016 peste-des-petits-ruminants virus (PPRV) has been renamed as Small ruminant morbillivirus (in italics), more information at https://talk.ictvonline.org//taxonomy/p/taxonomy-history?taxnode_id=201851619; line 57 “Measles virus” had this name up to 2015. Since 2016 it has been renamed as Measles morbillivirus. Please correct. More information at https://talk.ictvonline.org//taxonomy/p/taxonomy-history?taxnode_id=201851616
ANSWER:
- Line 8: Morbillivirus were written in italics - Line 31: The text has been modified. However, concerning PPRV, we believe that in the current context of global eradication of PPR, it is not a good idea to change the name of the causative agent, as it may confuse communication and on-going efforts of many stakeholders, field actors, and non-specialists involved in the campaign. Therefore, we changed the text in order to explain why we retained the name PPRV in the manuscript.
2.in the text there are many double spaces among consecutive words and some typing mistakes are present, e.g. Line 12: “strain, please check if “,” is needed; line 28: “PPP” probably should be replaced with PPR, line 62: “[11][12]” should be [11,12], line 66: “characterized[13]” a space should be inserted; line 93: “strains[19]” a space should be inserted; line 112: “cell line, (ATCC CCL-226, USA)” should be “cell line (ATCC CCL-226, USA),”; line 138 “France).The” a space should be added after point; line 180: “highlight? Evolution” is “?” correct? Line 182: “was” or “were”?; line 199: “phenotype?” is correct?
ANSWER:
- Line 12: The comma after the word "strain" has been deleted.
- Line 28: PPP has been replaced by PPR
- Line 62:"[11],[12]" has been replaced by"[11,12]".
- Line 66: A space has been inserted between "characterized" and"[13]".
- Line 93: A space has been inserted between "strains" and"[19]".
- Line 112: "cell line, (ATCC CCL-226, USA)" has been replaced by "cell line (ATCC CCL-226, USA)".
- Line 138: A space has been inserted between "France)" and "The".
- Line 180: The question mark has been removed
- Line 182: The words "was" have been replaced by "was".
- line 199: The question marks have been deleted
3.line 71: “Producers are typically advised not to go further than five cell passages of the vaccine seed during production to avoid risks of reduced vaccine efficacy and of eventual reversal to virulence.” if this sentence is supported by references, please insert them.
ANSWER : The following reference have been added : OIE. Peste des petits ruminants (infection par le virus de la peste des petits ruminants). In Manuel des tests de diagnostic et des vaccins pour les animaux terrestres 2019. (http://www.oie.int/fileadmin/Home/fr/Health_standards/tahm/3.07.09_PPR.pdf)
4.Line 105: I suggest to replace “(P89, 90, 99,130,149)” with (P89, P90, P99, P130, P149)
ANSWER: “(P89, 90, 99,130,149)” have been replaced by (P89, P90, P99, P130, P149)
5.Line 106: please check if “(passage 76 to 85)” should be changed with (passages 76 to 85)
ANSWER: “passage” have been modified by “passages”
6.Table 1 and Table 2: I suggest to mark the non-synonymus mutations that are described at lines 222-223.
ANSWER: The positions that host the non-synonymous mutations have been highlighted
6.Line 201: Please, could you add more information on the antibodies? Were they commercial? Please, insert references.
ANSWER: The following reference have been added for the NPPRV antibody : Libeau, G.; Lefevre, P.C. Comparison of rinderpest and peste des petits ruminants viruses using anti-nucleoprotein monoclonal antibodies. Vet. Microbiol. 1990, 25, 1–16. This antibody is not commercialized.
7.Line 247: I suggest to write P before each number or to write "at passages" and to write only numbers
ANSWER : A “P” was added before each passage number
8.Figure 3: Please insert a title to each x- and y-axis
ANSWER: The x-axis have been named NMDS axis 1 and the y-axis have been named NMDS axis 2
9.Figure 3: “(like other Ps… in these plots).” “Ps” is not clear
ANSWER: Ps… have been replaced by P…
10.Line 271: is the correct number Figure 4 instead of Figure 3?
ANSWER: The correct number of the figure “4” have been marked
11.Line 273 “morbilliviruses” italics is not required
ANSWER: Italics have been removed from “morbillivirus”
12.Line 294: (MeV) has already been defined at line 57, please delete
ANSWER: MeV have been deleted
13.Line 346-348: “Are there some reference supporting this sentence? Is it an observation done by the authors? In this latter case, I suggest to add these parts in the material and methods and in the results sections.
ANSWER: Material and methods part have been added for this sentence and the resulting alignment file will be given in supplemental material
Line 391: please insert details to reference number 3 (e.g. Internet web site address).
ANSWER: The internet site web address ( http://www.fao.org/3/a-av222e.pdf ) have been added to the reference